# The Impact of Milk Fat Globule Membrane with Exercise on Age-Related Degeneration of Neuromuscular Junctions

**DOI:** 10.3390/nu13072310

**Published:** 2021-07-05

**Authors:** Satoshi Sugita, Kotaro Tamura, Michiko Yano, Yoshihiko Minegishi, Noriyasu Ota

**Affiliations:** Biological Science Research, Kao Corporation, 2606 Akabane, Ichikai-machi, Haga-gun, Tochigi 321-3497, Japan; sugita.satoshi@kao.com (S.S.); tamura.kotaro@kao.com (K.T.); yano.michiko@kao.com (M.Y.); ota.noriyasu@kao.com (N.O.)

**Keywords:** aging, dietary ingredient, dying-back neuropathy, milk fat globule membrane, motor function, neuromuscular synapse, neuromuscular junction

## Abstract

Morphological changes in neuromuscular junctions (NMJs), which are synapses formed between α-motor neurons and skeletal muscle fibers, are considered to be important in age-related motor dysfunction. We have previously shown that the intake of dietary milk fat globule membrane (MFGM) combined with exercise attenuates age-related NMJ alterations in the early phase of aging. However, it is unclear whether the effect of MFGM with exercise on age-related NMJ alterations persists into old age, and whether intervention from old age is still effective when age-related changes in NMJs have already occurred. In this study, 6- or 18-month-old mice were treated with a 1% MFGM diet and daily running wheel exercise until 23 or 24 months of age, respectively. MFGM treatment with exercise was effective in suppressing the progression of age-related NMJ alterations in old age, and even after age-related changes in NMJs had already occurred. Moreover, the effect of MFGM intake with exercise was not restricted to NMJs but extended to the structure and function of peripheral nerves. This study demonstrates that MFGM intake with exercise may be a novel approach for improving motor function in the elderly by suppressing age-related NMJ alterations.

## 1. Introduction

As the world’s population ages, it is very important to attenuate the age-related decline in motor function in order to maintain quality of life [1,2]. The loss of muscle weight has been a focus of research because it is a characteristic change associated with aging [3,4,5]. However, longitudinal studies have shown that muscle weakness begins long before muscle atrophy occurs, suggesting that changes in the peripheral nervous system may precede muscle atrophy [6].

One of the neural factors that can result in muscle weakness is age-related dysfunction of the neuromuscular junction (NMJ), the synapse formed between α-motor neurons (α-MNs) and skeletal muscle fibers. In humans and rodents, it has been reported that the morphology of NMJs is altered with aging [7,8,9,10], and the changes in NMJs precede MN alterations [11]. In addition, these changes are associated with the functional impairment of the postsynaptic response of the NMJs [12]. Based on these observations, maintaining the integrity of NMJs may be an effective way to ameliorate age-related motor dysfunction.

Several food ingredients have been shown to have a potential effect on NMJs, including vitamin D, omega-3 fatty acids, and β-hydroxy-β-methylbutyrate; however, it is not clear whether they are effective for age-related NMJ degeneration [13]. Resveratrol has been shown to be effective against NMJ degeneration in old mice, but it was administered at 1 year of age, a time when NMJ degeneration has barely occurred [14]. The milk fat globule membrane (MFGM), a membrane that encloses milk lipid droplets, is a source of multiple bioactive compounds, including glycerophospholipids and sphingolipids, which are abundant in the nervous system [15]. We have previously shown that dietary MFGM intake combined with exercise improves physical performance and increases the expression of Dok-7 [16,17,18,19,20,21,22] which plays an essential role in NMJ formation [23,24]. Furthermore, we found that the intake of MFGM with exercise from a young age (6 months) attenuates the age-related morphological deterioration of NMJs in mice in the early phase of aging (at 14 months of age) [25]. These findings suggest that MFGM intake with exercise can improve motor function by suppressing age-related NMJ dysfunction. However, it is unclear whether the effect of MFGM with exercise on age-related NMJ alterations persists into old age, and whether intervention at an old age is still effective when age-related changes in NMJs have already occurred.

Here, we demonstrate that the effect of MFGM with exercise on NMJ alterations is not specific to middle-aged mice, but is maintained into old age. Interventions starting at the age of 18 months, when age-related changes in NMJs have already occurred [8], had an inhibitory effect on NMJ alterations. Moreover, we found that the effect of MFGM with exercise was not restricted to NMJs, but extended to the structure and function of nerves. This suggests that MFGM with exercise has the potential to be a new approach for improving motor function in the elderly to suppress age-related NMJ alterations.

## 2. Materials and Methods

### 2.1. Animals

BALB/c and C57BL/6J mice were purchased from Charles River Japan and maintained at 23 ± 2 °C under a 12 h light–dark cycle. All animal experiments were approved by the Animal Care Committee of Kao Corporation and were performed in accordance with the committee’s Guidelines for the Care and Use of Laboratory Animals. 

### 2.2. Materials and Experimental Diets

MFGM was prepared from buttermilk by filtration and centrifugation [25]. The composition of MFGM was as follows: 53.4% protein, 25.2% fat, 13.4% carbohydrate, 5.5% ash, 2.5% moisture, and 20.4% phospholipids (5.69% phosphatidylcholine, 5.69% phosphatidylethanolamine, 1.76% phosphatidylinositol, 2.03% phosphatidylserine, 3.72% sphingomyelin, and other phospholipids). Mice were allowed ad libitum access to water and a powder diet during the experiments. The control diet contained 10% fat (*w*/*w*), 20% casein, 55.5% potato starch, 8.1% cellulose, 4% minerals, 2.2% vitamins, and 0.2% methionine. In the MFGM group, 1% MFGM was added to the control diet.

### 2.3. Experimental Design 

We performed two experiments in this study (Figure 1). In Experiment 1, we conducted an intervention from a young age (6 months). In Experiment 2, we conducted an intervention from an old age (18 months) when age-related NMJ changes had already occurred [8].

#### 2.3.1. Experiment 1

Male BALB/c mice (6 months old) were fed a laboratory chow (CE-2, CLEA Japan) and allowed to drink freely for 1 week to acclimate to the environment. After acclimatization, a rotarod test was performed to measure motor coordination. The mice were then divided into the following four groups so that body weight and rotarod score were comparable: control diet (CONT), MFGM diet (MFGM), control diet with exercise (EX-CONT), and MFGM diet with exercise (EX-MFGM). All mice were individually housed in plastic cages, and the exercise group was housed in plastic cages with running wheels (SW-15 mg, MerckQuest). Male BALB/c mice aged 6 months were used as young controls (YOUNG).

#### 2.3.2. Experiment 2

Male C57BL/6J mice (18 months old) were divided into the following three groups so that body weight was comparable: control diet (Cont), control diet with exercise (Ex-Cont), and MFGM diet with exercise (Ex-MFGM). All mice were individually housed in plastic cages, and the exercise group was housed in plastic cages with running wheels (SW-15 mg, MerckQuest). Male C57BL/6J mice (10 months old) were used as young controls (Young).

### 2.4. Rotarod Test

Mice were placed on a rotarod (MK-610A, Muromachi Kikai, Tokyo, Japan). The time that the mice were able to spend on the rotarod was recorded. The rotarod was set to an acceleration program that increased the speed from 4 rpm to 40 rpm within 5 min. Each mouse was tested three times per day for four consecutive days. When comparing the ability of mice to stay on the rotarod, we used the average of three trials on day 4.

### 2.5. NMJ Analysis

Mice were anesthetized with isoflurane (Pfizer, New York, NY, USA) and perfused transcardially with phosphate-buffered saline (PBS), followed by 4% paraformaldehyde in PBS (FUJIFILM Wako Pure Chemical Corporation, Osaka, Japan). The extensor digitorum longus muscle was isolated. After incubation in blocking solution (1% Triton X-100, Sigma; 3% bovine serum albumin, Sigma; 5% goat serum, Vector Laboratories, in PBS) for 1 h at room temperature (RT), whole muscles were incubated with the primary antibody against synapsin I (#5297S, Cell Signaling, Danvers, MA, USA, 1:400) in blocking solution overnight at 4 °C. The muscles were washed three times with PBS-T (0.05% Tween 20, Bio-Rad) and incubated with Alexa 488-conjugated fluorescently tagged α-bungarotoxin (#B13422, Life Technologies, Carlsbad, CA, USA, 1:400) and the secondary antibody (Alexa 555 or 594 anti-rabbit IgG, Life Technologies, 1:400) for 2 h at RT. After washing with PBS-T, the muscles were whole-mounted onto slides using VECTASHIELD Hard-Set Mounting Medium (Vector Laboratories, Burlingame, CA, USA).

To analyze the structural features of NMJs, digital images were acquired with a confocal microscope (LSM880, Carl Zeiss, Jena, Germany), and maximum intensity projections of confocal stacks were created using ZEN software Blue version (Carl Zeiss). We analyzed the structural features according to the criteria previously described in [8,26]. Briefly, fragmented acetylcholine receptors (AChRs) are defined as five or more AChR clusters in small islands with a round shape. Denervated NMJs describe postsynaptic sites not fully or partially apposed by the nerve terminal. At least 50 NMJs per mouse were analyzed.

### 2.6. Western Blot Analysis

The gastrocnemius muscle was isolated immediately following perfusion with PBS and stored at −80 °C before use. The muscles were homogenized using a hand homogenizer (Microtech, Funabashi, Japan) in ice-cold lysis buffer (CelLytic Mammalian Tissue Lysis/Extraction Reagent, Sigma, St. Louis, MO, USA) supplemented with a protease inhibitor cocktail (cOmplete, Roche, Basel, Switzerland). After centrifugation of the muscle mixtures at 13,500× *g* for 15 min at 4 °C, protein concentrations of the supernatants were measured using a BCA protein assay (Pierce, Waltham, MA, USA). The lysates were separated on an SDS-polyacrylamide gel (Bio-Rad, Hercules, CA, USA) and transferred onto polyvinylidene fluoride membranes (Bio-Rad). The membranes were then blocked with PVDF Blocking Reagent (Toyobo, Osaka, Japan) for 1 h at RT. Subsequently, the membranes were incubated with appropriate primary antibodies against Dok-7 (#sc-55169, Santa Cruz Biotechnology, Dallas, TX, USA, 1:1000) and GAPDH (#2118S, Cell Signaling, 1:5000) in immunoreaction enhancer solution (Can Get Signal, Toyobo) overnight at 4 °C. Goat (#sc-2354, Santa Cruz) or rabbit (#7074S, Cell Signaling) secondary antibodies conjugated with horseradish peroxidase were used for detection via enhanced chemiluminescence using ECL Prime (GE Healthcare, Chicago, IL, USA). Blots were acquired and analyzed using ChemiDoc Touch (Bio-Rad). The data were normalized to the background and loading controls.

### 2.7. Nerve Histology

To visualize nerve structures, peroneal nerves were isolated immediately following perfusion with PBS, pre-fixed with 2.5% glutaraldehyde overnight at 4 °C, and then post-fixed with 1% osmium tetroxide for 2 h at 4 °C. After embedding in EPON resin (TAAB Laboratories), specimens were cut at 1.5 µm thickness and stained with 0.5% toluidine blue. Bright-field images of the sections were acquired using an all-in-one fluorescence microscope (BZ-X700, Keyence, Osaka, Japan) at a magnification of 40×. To evaluate the morphology of peroneal nerves, the total numbers of axons were manually counted. Structural myelin abnormalities such as myelin inclusions, myelin outfoldings, and tomacula were also evaluated as previously described [27]. The g-ratio (the ratio between the inner and outer diameters of the myelin sheath) was determined on approximately 200 axons per section, randomly picked with the help of a grid (the vertex of a 500 μm^2^ square) using ImageJ software (National Institutes of Health).

For neurofilament immunostaining, mice were anesthetized with isoflurane (Pfizer) and perfused transcardially with phosphate-buffered saline (PBS), followed by 4% paraformaldehyde in PBS (Wako). The tibial nerve was isolated and paraffin-embedded. The sections of 3 μm were deparaffinized and activated with Target Retrieval Solution, pH 9 (3-in-1, Dako, Santa Clara, CA, USA) warmed to 97 °C for 23 min and then left at room temperature for 30 min. After incubation in blocking solution (0.1% Triton X-100, Sigma; 3% bovine serum albumin, Sigma; 5% donkey serum, in PBS) for 1 h at room temperature, the sections were incubated with the primary antibody against neurofilament (2F11, Dako) in blocking solution overnight at 4 °C. The sections were washed three times with PBS-T (0.05% Tween 20, Bio-Rad) and incubated with the secondary antibody (Alexa 488 anti-goat IgG, Life Technologies, 1:500) for 3 h at RT. After washing with PBS-T, the sections were mounted using VECTASHIELD Hard-Set Mounting Medium (Vector Laboratories). After imaging with an all-in-one fluorescence microscope (BZ-X700, Keyence), neurofilaments expressing nerves were counted.

### 2.8. Spinal Cord Histology

Immunostaining of the spinal cord was performed as described previously [28]. Briefly, the vertebral column containing the spinal cord was dissected and post-fixed in 4% paraformaldehyde overnight at 4 °C. The lower lumbar segments (L3–L5) of the spinal cords were isolated and impregnated with 30% sucrose in PBS for over 3 days at 4 °C, followed by embedding in OCT compound (Sakura Finetek Japan, Osaka, Japan) and stored at −80 °C. The spinal cords were sectioned to a thickness of 30 μm using a cryostat (Leica, Wetzlar, Germany). Six sections per mouse (approximately every tenth section) were collected and subjected to immunolabeling in a free-floating manner. The sections were incubated with primary antibodies against NeuN (#MAB377, EMD Millipore, Burlington, MA, USA 1:1000) and cleaved caspase-3 (#9664, Cell Signaling, 1:500) with a diluted solution (1% normal donkey serum, 0.3% Triton X-100, 0.25% λ-carrageenan, 0.01% NaN_3_ in PBS) on a shaker at 4 °C for 3 days. The sections were washed in PBS-T three times and incubated with appropriate secondary antibodies (Alexa 488-conjugated anti-rabbit and Alexa 594-conjugated anti-mouse IgG, Life Technologies, 1:1000) with the diluted solution for 1 h at RT. The sections were then washed three times in PBS-T and mounted onto slides using VECTASHIELD Hard-Set Mounting Medium. The images were acquired with a fluorescence microscope (BZ-X700, Keyence) by Z-stacks with 10–20 μm depth from the surface and 1 or 2 μm intervals. NeuN-positive cells located in the ventral horn (lamina IV) were defined as motor neurons (MNs). Only MNs showing a large nucleus and a healthy appearance were considered in the following evaluations, and approximately 50–70 MNs were evaluated per mouse. For caspase-3 analysis, caspase-3-positive MNs were manually tagged, and the percentage of these MNs was calculated. 

### 2.9. Motor Nerve Conduction Velocity

Mice were anesthetized with isoflurane (Pfizer) and placed on a heat-retaining pad (39DP, Muromachi Kikai, Tokyo, Japan) to maintain body temperature at 37 °C. Motor nerve conduction velocity (MNCV) was determined using established methods [29,30,31]. In brief, the right sciatic-tibial nerve was electrically stimulated by needle electrodes (MEB9402-MB, Nihon Kohden, Tokyo, Japan) at the ankle and similarly at the sciatic notch, and M-waves were recorded from the second interosseous muscle of the foot. The distance between the two stimulation sites was divided by the latency difference.

### 2.10. Statistical Analyses

All values are presented as the mean ± standard error (SE). To compare the means of more than two groups, we used a one-way analysis of variance, followed by Tukey’s test (SPSS software version 24, IBM, Armonk, NY, USA). Statistical significance was set at *p* < 0.05.

## 3. Results

### 3.1. Experiment 1

#### 3.1.1. Effects of MFGM with Exercise from a Young Age on Body and Muscle Weight

To determine whether the inhibitory effects of MFGM with exercise on age-related changes in motor function and NMJs observed in middle-aged animals [25] were maintained into old age, running wheel exercise and MFGM loading were performed from the age of 6 months. Body and muscle weights at 23 months of age are presented in Table 1. The weight of the extensor digitorum longus muscle in the CONT, MFGM, and EX-CONT groups was significantly smaller than that in the YOUNG group, but there were no significant differences in body, extensor digitorum longus muscle, soleus muscle, tibialis anterior muscle, and gastrocnemius muscle weight among intervention groups.

#### 3.1.2. Effects of MFGM with Exercise from a Young Age on Motor Function

We performed a rotarod test to evaluate motor coordination. There was no significant difference in time on the rotarod in the EX-CONT group compared to the CONT group, but the time on the rotarod in the EX-MFGM group was significantly prolonged compared to that in the CONT group (Figure 2). These results suggest that a combination of MFGM and exercise can prevent age-related motor function decline in old age.

#### 3.1.3. Effect of MFGM with Exercise from a Young Age on Age-Related NMJ Alterations

We analyzed the morphology of NMJs. In the YOUNG group, NMJs had a pretzel-like shape, and AChRs (which were aggregated in the postsynaptic membrane) merged almost completely with axons (Figure 3A). In contrast to the YOUNG group, AChRs in the CONT group were often fragmented into small islands (Figure 3A,B), and AChR clusters on some muscle fibers were not apposed by an axon (Figure 3A,C). Compared to the CONT group, there was no significant change in the frequencies of the fragmentation of AChR aggregates and of denervated NMJs in the EX-CONT group, whereas there was a trend for a decrease in the percentage of fragmented AChRs (*p* = 0.07) and a significant decrease in the percentage of denervated NMJs in the EX-MFGM group (Figure 3A–C). These results suggest that the combination of MFGM intake and exercise attenuates age-related NMJ alterations.

### 3.2. Experiment 2 

#### 3.2.1. Effect of MFGM with Exercise from an Old Age on Age-Related NMJ Alterations

Next, we examined whether these interventions starting at 18 months of age (when age-related morphological changes in NMJs are more pronounced [8]) are effective in reducing age-related changes in NMJs. In the Cont group, the rates of fragmented AChRs and denervated NMJs were increased compared to those in the Young group (Figure 4A–C). In the Ex-Cont group, the percentage of fragmented AChRs was significantly reduced compared to that in the Cont group, but there was no significant difference in the percentage of denervated NMJs. By contrast, in the Ex-MFGM group, the percentage of fragmented AChRs and the percentage of denervated NMJs were significantly reduced compared to those in the Cont group.

Since the combination of MFGM intake and exercise from a young age has been shown to attenuate the age-related decline in the expression of Dok-7, which promotes NMJ formation [22,23,24,32,33], we examined whether MFGM with exercise from an old age affects the expression of Dok-7. The expression of Dok-7 protein was significantly higher in the Ex-MFGM group than in the Cont group (Appendix A).

#### 3.2.2. Effect of MFGM with Exercise from an Old Age on Nerve Structure and Function

In the NMJ analysis, compared to the exercise regimen, the effect of MFGM intake combined with exercise was more pronounced on presynaptic features (i.e., denervated NMJs; Figure 3C and Figure 4C). In addition, the concept of “dying-back neuropathy” has been proposed [11], in which changes in the NMJs cause neurodegeneration. Therefore, we examined whether a combination of MFGM intake and exercise could inhibit age-related neurodegeneration. 

In the Cont group, the number of nerve axons was reduced compared to that in the Young group (Figure 5A,B,H). The percentage of abnormal myelin morphologies such as myelin inclusions, outfoldings, and tomacular structures (Figure 5E–G) was also significantly higher than that in the Young group (Figure 5I). In addition, there was an increase in the g-ratio, an indicator of myelin thickness, compared to the Young group (Figure 5J). These structural changes in the nerves were associated with a decline in motor nerve conduction velocity (Figure 5K).

Exercise partially attenuated the age-related loss of nerve axons and increased the g-ratio (Figure 5H,J), but had no effect on the proportion of abnormal myelin or motor nerve conduction velocity (Figure 5I,K). Furthermore, the number of nerve axons in the MFGM group, which was loaded from a young age, did not change compared to the CONT group (Appendix A).

By contrast, the combination of MFGM with exercise almost completely suppressed the age-related decrease in the number of nerves and the age-related increase in myelin abnormalities and g-ratio (Figure 5H–J). In addition, the age-related decrease in motor nerve conduction velocity was suppressed by the combination of MFGM and exercise (Figure 5K). These results indicate that the combination of MFGM with exercise attenuates age-related changes in nerve structure and function.

Because axon loss is associated with motor nerve death, we examined the activation of caspase-3, an indicator of cell death, in the soma of MNs (Appendix A). We found an increase in cleaved caspase-3 in the Cont group compared to that in the Young group, and the combination of MFGM with exercise suppressed the age-related increase in caspase-3 (Appendix A).

## 4. Discussion

We demonstrated that the effects of MFGM intake combined with exercise on NMJ alterations are not only specific to middle-aged mice but also maintained into old age. Previously, we showed that the combination of MFGM intake with exercise suppressed NMJ morphological changes at 14 months of age, but severe NMJ abnormalities were rare at this time [25]. In addition, it is known that the morphology of NMJs in mice changes markedly from 18 months of age, and fragmentation of approximately 80% of AChRs occurs by the age of 24 months. Given the above, this study shows that MFGM intake combined with exercise attenuates age-related morphological changes in NMJs.

In the present study, we started the intervention of MFGM intake combined with exercise at the age of 18 months, a time at which age-related changes in NMJs had already occurred [8]. The age of 18–24 months in C57BL/6J mice is considered to be “old age”, equivalent to the age of 56–69 years in humans [34]. In our previous studies on elderly people over 60 years old, we reported that MFGM intake with exercise improved motor function and increased muscle fiber conduction velocity [17], suggesting the possibility of improving NMJ function. These findings suggest that MFGM intake combined with exercise has the potential to be a new approach for improving motor function in the elderly by suppressing age-related NMJ alterations.

Exercise is widely recognized as one of the most effective ways to improve motor function, and has also been shown to inhibit age-related NMJ degeneration [8,35]. In the present study, exercise decreased the number of fragmented AChRs. Conversely, there was no change in the percentage of denervated NMJs. Consistent with these results, previous reports have shown that the inhibitory effect of exercise on NMJ degeneration is stronger in postsynaptic features (e.g., fragmented AChRs) and weaker in presynaptic features (e.g., denervated NMJs) [8]. Although AChR fragmentation is a characteristic morphological change that increases with age, it has been suggested that this does not necessarily reflect NMJ dysfunction, but rather a structural adaptation to maintain neuromuscular transmission [36,37]. Indeed, it has been shown that in the mouse diaphragm, increased fragmentation of AChRs does not decrease the combined muscle action potential, but rather increases it [38]. By contrast, denervated NMJs, in which motor nerve endings are withdrawn from AChRs, have been shown to correlate with age-related motor function decline [25]. In addition, denervated NMJs are a characteristic feature of amyotrophic lateral sclerosis (ALS), a neurodegenerative disease [26,39]. In ALS, age-related physiological changes such as muscle atrophy, muscle weakness, and associated motor dysfunction are accelerated, suggesting that the amelioration of NMJ denervation may be a therapeutic target for age-related motor dysfunction [40,41]. In the present study, the combination of MFGM intake and exercise not only suppressed the occurrence of fragmented AChRs but also reduced the percentage of denervated NMJs, which cannot be achieved by exercise alone, indicating the significance of MFGM intake.

We found that the effect of MFGM with exercise was not restricted to NMJs but extended to the structure and function of nerves. In humans and rodents, aging has been shown to cause nerve morphological changes such as a decrease in the number of nerves and an increase in abnormal myelin morphology [27,42,43,44,45], as well as a decrease in nerve conduction velocity [46,47,48,49,50]. It has also been reported that exercise has no obvious effect on age-related changes [8,51]. Similarly, in the present study, the effects of exercise on age-related nerve structure changes were limited and did not lead to improvements in neural function. By contrast, the concomitant use of MFGM had a marked inhibitory effect on age-related changes in nerve structure, and on the reduction of nerve function. Several studies have shown that MFGM promotes the development of hippocampal neurons in weaned rats [52] and improves cognitive scores in infants [53], which may contribute to neurodevelopment. This study provides evidence of the effects of MFGM with exercise on age-related neural morphological changes and functional decline, and suggests that the intake of MFGM with exercise may be a new method to attenuate neural dysfunction in the elderly.

Although the cause of age-related neural dysfunction is not fully understood, the concept of “dying-back neuropathy” is an attractive hypothesis, in which NMJ degeneration is the starting point and affects the nervous system [11]. In the current study, the combination of MFGM intake with exercise extended its beneficial effects to the nerves in addition to the NMJs, but it is still unclear whether these effects act directly on the nervous system or secondarily through the NMJs. It is possible that the sphingomyelin contained in MFGM is supplied directly to nerves as a resource since it is a component of the myelin sheath. In one report, the administration of an inhibitor of sphingolipid synthesis in developing rats resulted in a loss of myelin sheath thickness, which was reversed by treatment with sphingomyelin [54]. By contrast, in the current study MFGM supplementation increased the protein expression of Dok-7 in muscles, which is consistent with the results of our previous studies [20,22]. We recently showed that the overexpression of Dok-7 in skeletal muscles ameliorated age-related NMJ denervation [32], and therefore MFGM may have secondary effects on the nervous system via muscles and NMJs. Further studies are needed to clarify whether MFGM directly affects the nervous system.

This study has a limitation that should be considered. Our findings on the effects of MFGM intake combined with exercise on NMJ and nerve function were only analyzed in male mice. It has been reported that there are no gender differences in the age-related NMJ alterations [55]. In addition, several lines of evidence have accumulated indicating that MFGM has beneficial effects in both sexes [17,19,56]. However, further studies are needed to elucidate whether gender differences exist.

In conclusion, the effect of MFGM with exercise on NMJ alterations is not limited to middle-aged animals but is also sustained in old age, and even if interventions start at a time when age-related changes in NMJs have already occurred. Furthermore, we found that the effects of MFGM with exercise are not specific to NMJs but extend to the structure and function of nerves. This study provides evidence that MFGM with exercise may be a useful approach to prevent age-related changes in NMJs and improve motor function in the elderly.

## Figures and Tables

**Figure 1 nutrients-13-02310-f001:**
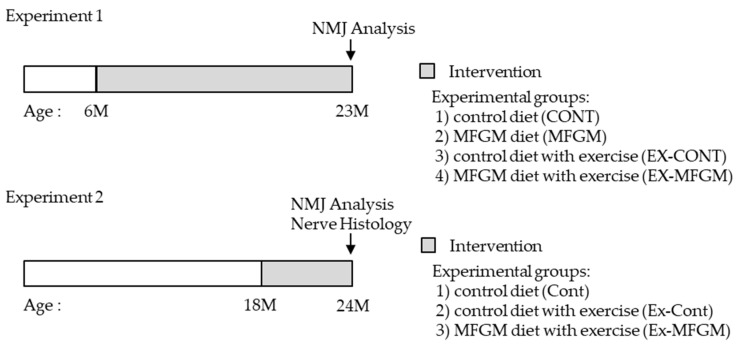
Timelines of the two experiments. MFGM, milk fat globule membrane; NMJ: neuromuscular junction.

**Figure 2 nutrients-13-02310-f002:**
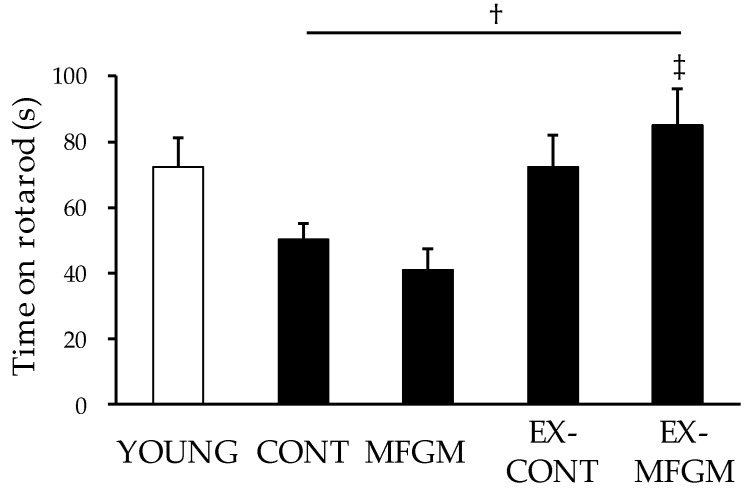
The effect of MFGM with exercise on motor function. The rotarod test was performed in 17-month-old BALB/c mice. Mice were placed on the rotarod accelerating from 4 rpm to 40 rpm. Values are means ± SE (*n* = 8–12 in each group). † *p* < 0.05 vs. CONT group, ‡ *p* < 0.05 vs. MFGM diet group by Tukey’s test. MFGM, milk fat globule membrane. YOUNG, young group; CONT, control diet group; MFGM, milk fat globule membrane diet group; EX-CONT, control diet with exercise group; EX-MFGM, milk fat globule membrane diet with exercise group.

**Figure 3 nutrients-13-02310-f003:**
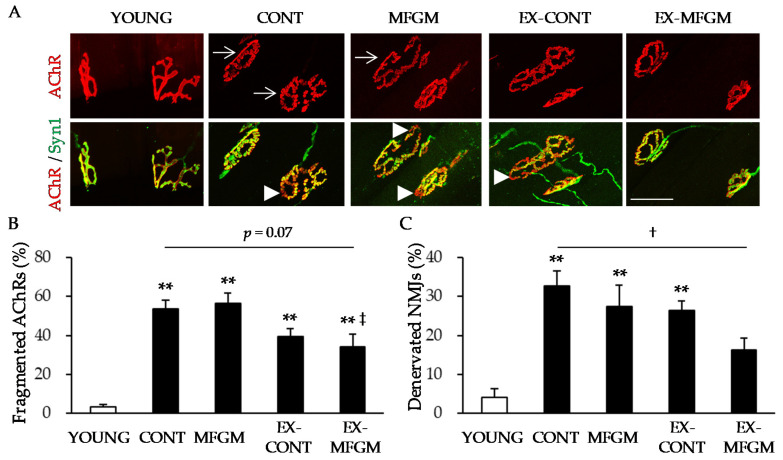
The effect of MFGM supplementation with exercise from a young age on age-related NMJ alterations. (**A**) Representative NMJ images at 6 months old (YOUNG) and 23 months old (CONT, MFGM, EX-CONT, EX-MFGM) are shown. Postsynaptic AChRs were stained with fBTX (red), and presynaptic nerve terminals were stained with antibodies against synapsin I (green). Arrows show fragmented AChRs, arrowheads denervated NMJs. Scale bar: 50 µm. The percentages of fragmented AChRs (**B**) and denervated NMJs (**C**) were determined. Values are presented as the mean ± SE (*n* = 4–5 in each group). ** *p* < 0.01 vs. YOUNG group, † *p* < 0.05 vs. CONT group, ‡ *p* < 0.05 vs. MFGM diet group by Tukey’s test. AChR, acetylcholine receptor; fBTX, fluorescently tagged α-bungarotoxin; MFGM, milk fat globule membrane; NMJ, neuromuscular junction; Syn1, synapsin I; YOUNG, young group; CONT, control diet group; MFGM, milk fat globule membrane diet group; EX-CONT, control diet with exercise group; EX-MFGM, milk fat globule membrane diet with exercise group.

**Figure 4 nutrients-13-02310-f004:**
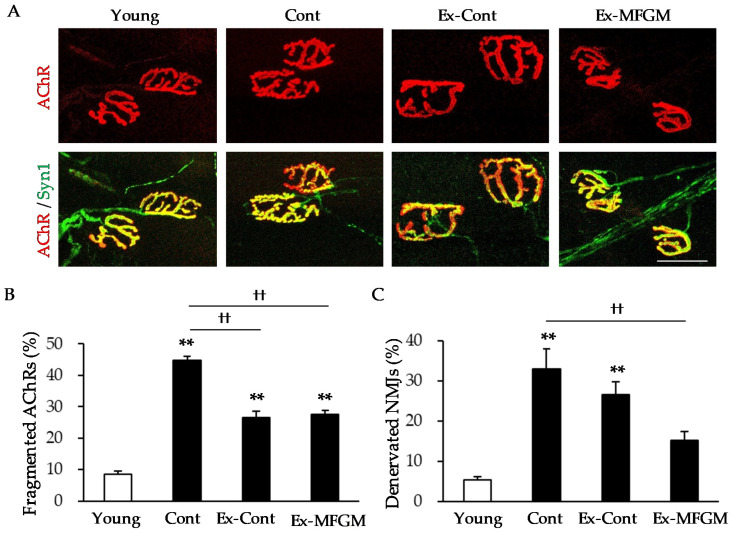
The effect of MFGM intake with exercise from an old age on age-related NMJ alterations. (**A**) Representative NMJ images at 10 months old (Young) and 24 months old (Cont, Ex-Cont, Ex-MFGM) are shown. Postsynaptic AChRs were stained with fBTX (red), and presynaptic nerve terminals were stained with antibodies against synapsin I (green). Scale bar: 50 µm. The percentages of fragmented AChRs (**B**) and denervated NMJs (**C**) were calculated. Values are presented as the mean ± SE (*n* = 4–7 in each group). ** *p* < 0.01 vs. Young group; †† *p* < 0.01 vs. Cont group by Tukey’s test. AChR, acetylcholine receptor; fBTX, fluorescently tagged α-bungarotoxin; MFGM, milk fat globule membrane; NMJ, neuromuscular junction; Syn1, synapsin I; Young, young group; Cont, control diet group; Ex-Cont, control diet with exercise group; Ex-MFGM, milk fat globule membrane diet with exercise group.

**Figure 5 nutrients-13-02310-f005:**
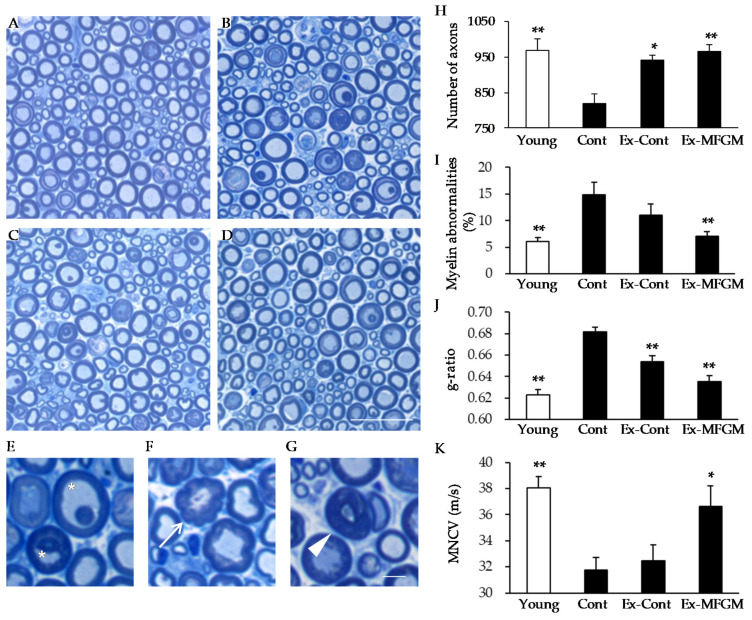
Effect of MFGM intake with exercise from an old age on nerve structure and function. (**A**–**G**) Semi-thin cross-sections of the peroneal nerve stained with toluidine blue. Low-magnification images of nerves at 10 months old (Young; (**A**)) and 24 months old (Cont, (**B**); Ex-Cont, (**C**); Ex-MFGM, (**D**)) are shown. High-magnification images of nerves are shown in (**E**–**G**). Age-related alterations included myelin inclusion (asterisk) (**E**), myelin outfoldings (arrow) (**F**), and tomacular structures (arrowhead) (**G**). (**H**) Number of axons present in the peroneal nerve. (**I**) The percentages of myelin abnormalities comprising myelin inclusions, myelin outfoldings, and tomacular structures were determined. (**J**) The ratio of nerve fiber diameter divided by axon diameter (g-ratio) was calculated. (**K**) The motor nerve conduction velocity was analyzed. Values are presented as the mean ± SE (*n* = 5–9 in each group). * *p* < 0.05, ** *p* < 0.01 vs. Young group. Scale bars: 30 µm (**A**–**D**) and 10 µm (**E**–**G**). MFGM, milk fat globule membrane; MNCV, motor nerve conduction velocity; Young, young group; Cont, control diet group; Ex-Cont, control diet with exercise group; Ex-MFGM, milk fat globule membrane diet with exercise group.

**Table 1 nutrients-13-02310-t001:** Body and muscle weights of BALB/c mice.

Group	YOUNG	CONT	MFGM	EX-CONT	EX-MFGM
Body weight (g)	30.6 ± 0.6	34.8 ± 1.1	36.5 ± 1.3 *	33.9 ± 0.9	36.0 ± 1.7 *
EDL muscle (mg)	14.6 ± 0.5	12.1 ± 0.4 *	11.7 ± 0.7 **	12.1 ± 0.5 *	12.7 ± 0.2
Soleus muscle (mg)	10.6 ± 1.3	10.1 ± 0.6	10.0 ± 0.4	13.0 ± 0.7	12.7 ± 0.9
TA muscle (mg)	53.3 ± 2.5	53.3 ± 2.6	50.2 ± 4.2	51.4 ± 3.4	54.0 ± 1.5
Gastro muscle (mg)	143.3 ± 4.7	127.3 ± 5.5	127.3 ± 11.4	134.7 ± 1.5	141.2 ± 5.5

Parameters were measured at the age of 6 months in the YOUNG group and at the age of 23 months in the CONT, MFGM, EX-CONT, and EX-MFGM groups. Values are shown as the mean ± SE (*n* = 3–5 in each group). * *p* < 0.05, ** *p* < 0.01 vs. YOUNG group by Tukey’s test. EDL, extensor digitorum longus; Gastro, gastrocnemius; TA, tibialis anterior; YOUNG, young group; CONT, control diet group; MFGM, milk fat globule membrane diet group; EX-CONT, control diet with exercise group; EX-MFGM, milk fat globule membrane diet with exercise group.

## Data Availability

All data are contained within the article or Appendix A.

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
