# Peer review of "The Impact of Milk Fat Globule Membrane with Exercise on Age-Related Degeneration of Neuromuscular Junctions"

_nutrients, 2021, doi:10.3390/nu13072310_

Round 1

Reviewer 1 Report

Dear Authors,

This is a solid paper and I think will make a great contribution to our knowledge.  I have three questions though:

  1. Why only male mice? Is this phenomenon only detected in males?
  2. A rotarod test seems good for testing stamina and balance, but could you also test strength?
  3. What test forelimb muscles, or upper hindlimb?  Does this only occur in lower hindlimb muscles?

Best wishes on future work.

Author Response

We would like to thank you for their helpful comments. In response, we have made significant changes to the manuscript. We have included a limitation that this study was conducted on male mice. Overall, the changes we made in this revised paper did not affect or alter our conclusions but rather helped us to demonstrate and explain them better. The details of the revisions are as follows:

Reviewer 2 Report

The manuscript entitled “The Impact of Milk Fat Globule Membrane with Exercise on Age-Related Degeneration of Neuromuscular Junctions” describes an interesting study to compare the combination of MFGM in the diet with exercise to either treatment alone in their abilities to impact degeneration of neuromuscular junctions. The general conclusion is that the combination of MFGM supplementation and exercise is superior to either treatment alone. While overall, this is a well written and clear study, there are several points that need to be addressed. My specific comments are as follows:

  • One thing that would help is a visual timeline of the two studies – I flipped around a bit to understand how long the treatments lasted based upon when they were administered and when the study was completed. The information is there but an image timeline for each of the two experiments would provide clarity to the reader.
  • It is concerning that only male mice were used – females should have been included. This is a major limitation of the study and needs to be discussed.
  • Re-consider the statistical assessment - the Dunnett’s test only informs us on significance between each treatment and the control. In figure 1 there appears to be no difference between young and the control and in several other assessments it is not clear whether there would be any significant difference between the treatment groups. ANOVA analyses across all cohorts would be more statistically informative.
  • MFGM only control groups need to be included in Figure 3 and Figure 4 to enable the stated conclusion of this study to be valid.

Author Response

We would like to thank you for your helpful comments. In response, we have made significant changes to the manuscript. We have included a limitation that this study was conducted on male mice. We have also included the visual timeline of experiments as Figure 1 in the Materials and Methods section. Overall, the changes we made in this revised paper did not affect or alter our conclusions but rather helped us to demonstrate and explain them better. The details of the revisions are as follows:

Round 2

Reviewer 2 Report

While most of my concerns have been addressed, I remain unconvinced that the chosen statistical method (alone) is appropriate. If this is one way you want to present the data, that is fine but I suggest adding analysis for significance across all cohorts as well. People sometimes use a different symbol to distinguish between the two significance indicators. When the combo is significantly different from the others under that analysis (even if not the case in every assay) it will strengthen your conclusions.

Author Response

We would like to thank you for your helpful comment. In response, we have made changes to the manuscript. We have modified the statistical methods for all data and compared the results across all groups. Overall, the changes we made in this revised paper did not affect or alter our conclusions but rather helped us to demonstrate and explain them better. 
